# The Copper(II)-Thiodiacetate (tda) Chelate as Efficient Receptor of N9-(2-Hydroxyethyl)Adenine (9heade): Synthesis, Molecular and Crystal Structures, Physical Properties and DFT Calculations of [Cu(tda)(9heade)(H_2_O)]·2H_2_O

**DOI:** 10.3390/molecules28155830

**Published:** 2023-08-02

**Authors:** Carmen Rosales-Martínez, Antonio Matilla-Hernádez, Duane Choquesillo-Lazarte, Antonio Frontera, Alfonso Castiñeiras, Juan Niclós-Gutiérrez

**Affiliations:** 1Department of Inorganic Chemistry, Faculty of Pharmacy, University of Granada, 18071 Granada, Spain; carmenromar@correo.ugr.es (C.R.-M.); amatilla@ugr.es (A.M.-H.); 2Laboratorio de Estudios Cristalográficos, IACT, CSIC-Universidad de Granada, Avda. de las Palmeras 4, Armilla, 18100 Granada, Spain; duane.choquesillo@csic.es; 3Departament de Química, Universitat de les Illes Balears, Crta. de Valldemossa km 7.5, 07122 Palma de Mallorca, Spain; toni.frontera@uib.es; 4Department of Inorganic Chemistry, Faculty of Pharmacy, University of Santiago de Compostela, 15782 Santiago de Compostela, Spain; alfonso.castineiras@usc.es

**Keywords:** copper(II), thiodiacetate chelate, synthetic adenine nucleoside, molecular and crystal structure, chelate-nucleoside recognition, water-mediated interligand interactions, physical properties, DFT calculations

## Abstract

Considering that Cu(tda) chelate (tda: dithioacetate) is a receptor for adenine and related 6-aminopurines, this study reports on the synthesis, molecular and crystal structures, thermal stability, spectral properties and DFT calculations related to [Cu(tda)(9heade)(H_2_O)]·2H_2_O (**1**) [9heade: N9-(2-hydroxyethyl)adenine]. Concerning the molecular recognition of (metal chelate)-(adenine synthetic nucleoside), **1** represents an unprecedented metal binding pattern (MBP) for 9heade. However, unprecedentedly, the Cu(tda)-9heade molecular recognition in **1** is essentially featured in the Cu-N1(9heade) bond, without any N6-H⋯O(carboxyl tda) interligand interaction. Nevertheless, N1 being the most basic donor for N9-substituted adenines, this Cu-N1 bond is now assisted by an O2–water-mediated interaction (N6-H⋯O2 and O2⋯Cu weak contact). Also, in the crystal packing, the O-H(ol) of 9heade interacts with its own adenine moiety as a result of an O3–water-mediated interaction (O(ol)-H⋯O3 plus O3-H36⋯π(adenine moiety)). Both water-mediated interactions seem to be responsible for serious alterations in the physical properties of crystalline or grounded samples. Density functional theory calculations were used to evaluate the interactions energetically. Moreover, the quantum theory of atoms-in-molecules (QTAIM), in combination with the noncovalent interaction plot (NCIPlot), was used to analyze the interactions and rationalize the existence and relative importance of hydrogen bonding, chalcogen bonding and π-stacking interactions. The novelty of this work resides in the discovery of a novel binding mode for N9-(2-hydroxyethyl)adenine. Moreover, the investigation of the important role of water in the solid state of **1** is also relevant, along with the chalcogen bonding interactions demonstrated by the density functional theory (DFT) study.

## 1. Introduction

From a broad point of view, chemical processes can be understood as a favorable consequence of the molecular recognition [1] between two or more reactants. This point of view implies that molecular recognition is always a phenomenon. The “favorable conditions” are determined by the two complementary facets of any chemical process: thermodynamics and kinetics. On this broad basis, we can consider a quasi-infinite number of objectives and experiences aimed at a better understanding of chemical reactions, addressing the factors that determine whether they are or not possible, and whether they can be finally accomplished. These arguments led many of us to pay attention to the contribution of the so-called “weak interactions” that accompany the formation of more robust chemical bonds. In this context, complex phenomena that emerge in the chemical and biological context [2] need to be discussed, from much simpler “model compounds” [3]. In past decades, we, as other researchers, started to work on mixed-ligand metal complexes formed by reactions between a simple metal salt or a metal chelate with N-bases of nucleic acids, nucleotides, natural or synthetic [4] nucleosides or nucleic bases (see, i.e., [5,6]). For the latter, efforts in preparing single crystals become essential because appropriate discussions of their structural results yield detailed information of both their own structures as well as how these compounds recognize themselves to build stable crystals, with or without the implication of solvents [7,8]. We recently became interested in the synthesis and molecular and/or crystal structures of mixed-ligand metal complexes with a synthetic purine nucleoside, as source of the metal binding patterns (MBD) of these latter ligands and of the analysis of the interligand interactions contributing to these ternary complexes at molecular and supramolecular levels. In this regard, the main support involves compounds having acyclovir (acv) or N9-(2-hydroxyethyl)adenine (9heade) as coligands (see Figure 1).

As a whole, the data in Table 1 reveal that all previous reports, with crystallographic support, are more numerous for acv than for 9heade. In addition, it is noteworthy that the N9-substituents in both synthetic nucleosides preclude their own tautomeric possibilities for the dissociable H(N9) atom of guanine or adenine. This seems to be the main reason that N7 acts as a preferential heterocyclic donor atom of acv and 9heade. This role will cooperate with other N or O donors of both nucleosides. The maximum expression of this is provided by the acv-H anion (which coexists with molecular acv) in the polymer DIDJUY [9]. It should also be noted that, while O6-acv is involved as H-acceptor in intramolecular interligand H-bonds, the H-atoms of the exocyclic -N(6)H_2_ group of 9heade make it a hydrogen donor in such interactions. The hyperconjugation of the lone pair of electrons of this amino group with the aromatic purine moiety of the adenine strongly favors this H-donor role.

In clear contrast, only a few recent reports (nine metal complexes) involve 9heade as ligand, as summarized in Table 1. Only MBP-2 and occasionally MBP-3 are observed in these complexes. These patterns correspond to the cooperation of a metal(II)-N7 coordination bond, along with an appropriate interligand H-bonding interaction (N-H⋯O), as observed in the referred complexes (Table 1). In these complexes, the MII-N7(9heade) bond is cooperating with the (9heade)N(6)-H⋯O(coligands) bond. An unusual cooperation, in this kind of compounds, is twice featured in the tetrahedral complex of zinc(II) (Table 1) involving each Zn-N7(9heade) bond with each (9heade)N6-H··Cl interligand interaction. Both MBP2 and MBP3 are featured in the dinuclear complex molecule of glycylglicinate(2-) (Table 1).

The hydrogen bonding (HB) is the most studied and used noncovalent interaction in chemistry, biology and materials [15,16,17]. However, other forces like σ-hole interactions (e.g., halogen bonding [10]) involving elements of the p-block as electron acceptors (playing the role of the H-atom) are gaining attention among the scientific community, and specially in crystal engineering and supramolecular chemistry [10,18]. Depletion of electron density usually occurs on the extension of covalent bonds involving p-block elements, which are called σ-holes. Chalcogen bonding (ChB) has been recently recognized by the IUPAC [19] and defined as the “net attractive interaction between an electrophilic region associated with a chalcogen atom in a molecular entity and a nucleophilic region in another, or the same, molecular entity”. Several theoretical works have shown that the origin of ChB is mainly electrostatic [20], though the orbital donor–acceptor can also be very important (LP → σ*) [20]. The strength of the interaction depends on the polarizability of the Ch atom, increasingly for heavier Ch atoms [21]. Notably, it has been recently proved that the metal binding of the chalcogen atom increases its ability as an electron donor to participate in this kind of noncovalent interaction [22].

This work aims to report the synthesis, molecular and crystal structures of the title compound [Cu(tda)(9heade)(H_2_O)]·2H_2_O (1). Additionally, the study includes an investigation of its thermal stability, various spectral properties and theoretical calculations. The focus of the latter is on the relevance of σ-hole interactions involving the Cu-S (thioether atom) of the tda ligand along with other more conventional interactions like H-bonds and lone pair-π (lp–π) interactions. As mentioned earlier, this type of ChB interactions, where σ-hole donor atom is coordinated to transition metals, has been scarcely described and exploited in the literature [22] The ChBs interactions have been characterized energetically using DFT calculations, the quantum theory of atoms-in-molecules (QTAIM), molecular electrostatic potential (MEP) surfaces and the noncovalent interaction plot (NCIPlot) computational tools.

## 2. Results and Discussions

### 2.1. About the Strategy of Synthesis

The reported strategy for the synthesis of [Cu(tda)(9heade)(H_2_O]·2H_2_O (hereafter compound **1**) consists of a two-step reaction (see Section 4.1). First, green basic carbonate of copper(II), malachite, Cu_2_(CO_3_)(OH)_2_ reacts with H_2_tda in a molar ratio 1:2, corresponding to an equimolar Cu(II):tda ratio, in water, with CO_2_ being the only by-product. It is recommended to filter the resulting greenish solution without vacuum to remove, as far as possible, small amounts of unreacted malachite, which can act as crystallization nuclei (Figure 2). 

To the Cu(tda) aqueous solution, an equimolar amount of 9heade is slowly added with stirring. Mother liquors of **1** are filtered without vacuum on a crystallizer, which is covered by a plastic film to moderate the evaporation of the solvent. This procedure yields long greenish parallelepiped crystals, some of which are suitable for the X-ray analysis (Figure 1). These crystals were collected by successive filtrations (up to ~70% of yield) and were also used for thermal and spectral studies. Final fractions that may be contaminated with by-products, assessed using electron spin resonance (ESR) measurements, are discarded.

### 2.2. Molecular and Crystal Structure of Compound **1** and Their Relevant Significance for Molecular and Supra-Molecular Recognition

Relevant crystal data and structure refinement of [Cu(tda)(9heda)(H_2_O)]·2H_2_O (1) are summarized in Table 2. Coordination bond lengths and trans-angles for this tentative formula are reported as Appendix A (here after Appendix A. Data on H-bonding interactions are provided in Appendix A.

### 2.3. Molecular and Crystal Structures of Compound [Cu(tda)(9heade)(H_2_O)]·2H_2_O (1)

In a rather flexible visualisation, we can consider that the O2–water atom is at a distance of 3.06 Å from the metal, resulting a very weak Cu(1)⋯O(2) water contact (the sum of Van der Waals (VdW) radii of Cu (1.40 Å) and O (1.52 Å) is 2.92 Å). Indeed, it is well known that these VdW radii are only available for a limited number of elements and they are averaged values, affected by a certain error. From this point of view, coordination bond lengths and angles are reported in Table 3, and Figure 2 represents the crystal structure of the novel compound according to the formula [Cu(tda)(9heda)(H_2_O)]·2H_2_O in a very asymmetrically elongated octahedral coordination, type 4 + 1 + 1 *, where * denotes the above referred Cu⋯O2(aqua) weak contact (see discussion below). 

Ignoring the relevance of such very weak Cu⋯O2(aqua) contact, the copper(II) coordination in **1** can be understood as axially elongated square-based tetrahedral coordination, type 4 + 1, with the O1-aqua atom as distal donor, having the chemical formula [Cu(tda)(9heda)(H_2_O)]·2H_2_O (Figure 2).

In a simplified and more realistic chemical sense, compound **1** can be represented well by the formula [Cu(tda)(9heade)(H_2_O)], where the Cu(II) center is in a rather frequent elongated square-base pyramidal coordination environment, type 4 + 1 (Figure 3). 

The four basal donors in compound **1** are the O(4) and S(1) atoms from the tridentate tda chelator plus the O1-(aqua) and N21-(9heade) atoms. The distal donor atom in this case is O(8)-tda. Notably, the proximal Cu-O bond distances are always very close to 2 Å, making O(8) atom the considered distal donor atom in the 4 + 1 Cu(1) coordination, despite the Cu(1)-S(1) distance of 2.363 (1) Å. Consequently, the tridentate tda chelator exhibits a fac-SO + O(distal) conformation in compound **1**. This behavior has previously reported in three compounds [23,24,25] including [Cu(tda)(Hhyp)(H_2_O)]·2H_2_O (Hhyp = hypoxanthine) [24]. In this latter compound, the metal also exhibits an elongated square-based pyramidal coordination, where the N9-unsubstituted Hhyp binds the Cu(II) center through its most basic N9 donor atom in a trans configuration compared to the S-tda donor, and the O-aqua ligand occupies one of the four closest coordination sites. However, it should be noted that, in contrast, the O1-aqua ligand in compound **1** is trans compared to the O4-tda donor. In all three previously reported compounds, the Cu-S(tda) bond distance falls within a narrow range of 2.366–2.373 Å, with an average value of 2.369 Å, very close to the Cu-S(tda) bond of **1** (2.363 (1) Å, see Table 3). In clear contrast, eight known Cu(II)-tda compounds with fac-O_2_ + S(distal) conformation have longer Cu-S (distal) distances (ranging from 2.570 to 2.742 Å) averaging 2.648 Å [26,27,28,29,30]. This group of compounds includes the 3D-coordination polymer [Cu(µ_3_-tda)]_2_ [29] and seven ternary complexes, where the four closest coordination bonds consist of two O-tda donors along with two N-heterocyclic donors from two monodentate ligands (imidazole or 3,5-dimethyl-1H-pyrazole) or a α,α-diimine as a bidentate chelator (2,2′-bypiridine or various 1,10-phenantrolines). 

The tda ligand is also able to play non-chelating roles for Cu(II), without a Cu-S(thioether) bond, as documented for the bridging µ_2_-O,O-tda [23,28] or a µ_2_-O,O′,O″,O‴-tda [31] modes. However, no Cu(II) complex with the tridentate chelating tda in a mer-SO_2_ conformation has been reported so far! 

On the other hand, in compound **1**, 9heade displays an unprecedented MBP, using its most basic N-heterocyclic donor instead of its usually preferred N7 donor. The exocyclic -N(6)H_2_ group falls coplanar to the purine moiety because the N6-lone pair is delocalized by hyperconjugation with the aromatic purine moiety, resulting in a larger steric hindrance over N1 versus N7. A likely explanation is that, for 6-minopurines, the cooperation of the Cu-N7 bond with a N6-H⋯O(acceptor) represents a ring motif of seven atoms, whereas the cooperation of the Cu-N1 with a N6-H⋯O(acceptor) generates a ring motif of six atoms (also taking into account that the involved angles in these motifs are also relevant). In simpler words, the metal binding to the N7 donor of a 6-aminopurine is less hindered than the coordination to N1. Despite limited information on MBPs for 9heade, the µ_2_-N1,N7-bridging mode has been reported for a coordination polymer having cis-1,2-cyclobutanedicarboxylate(2−) anion as coligand [14]. However, this bridging mode is not observed in compound **1**, where the cooperation between the Cu-N1(9heade) with the O2–water-mediated H-bonding interaction represents an alternative favoring the copper(II) binding with the most basic N1-9heade. Note that the cooperation between a Cu-N7 bond and a N6-H⋯O(carboxylate) in 6-aminopurines generates a seven-membered ring motif, whereas the involvement of Cu-N1 instead of Cu-N6 reduces the dimension of such a ring motif to six atoms.

### 2.4. Physical Properties

The thermal stability, FT-IR, UV–Vis and RSE spectroscopies of compound **1** have been investigated, as detailed below.

The registration of the FT-IR spectrum reveals unusual behavior in the preparation of sample pellets with dry IR quality KBr, notably distinct when crystalline samples of **1** are grounded more and more (see Appendix A). Such behavior is minimized when the pellet is obtained with freshly prepared microcrystalline samples and minimally grounded with KBr (Appendix A). The spectrum indicates the presence of water/aqua, fac-SO + O(distal)-tda and 9heade, with the following wavenumber (cm^−1^) bands (those bands with contributions from two or more IR chromophores are not referred in this discussion, see Table 4). For water, ν_as_ 3446. For tda, ν_as_(CH_2_) 2952, 2925, ν_s_(CH_2_) 2884, ν_s_(COO) 1391, 1373 and ν(C-S) 647 (expected at 715–629, and rather weak because it is Raman active) [32,33]. For 9heade, ν(O-H) 3521, ν_as_(NH_2_) 3313, ν(C-H)_arom_ 2970, skeletal in purines 1686, 1571, δ(O-H)_ol_ 1339, ν(C-O)_ol_ 1060, out-of-plane π(C-H)_arom_ 876, 858 (usually 900–860). Relevant insights include the two symmetric stretching bands, ν_s_(COO), related to the fac-SO + O(distal) conformation of tda (instead of a band near 1385) as well as the loss of intensity of the absorption near 1625 for the grounded sample, where the contribution of the scissoring mode of water, δ(H_2_O), is overlapped with that of the δ(NH_2_) for 9heade. On the other hand, the reflectance spectrum obtained with a grounded sample of **1** shows an anomalously and poorly informative broad band centered at about 750 nm (see Appendix A).

The thermal stability of **1** was firstly studied by thermo-gravimetric analysis (TGA, r.t. −950 °C), with FT-IR identifications of the evolved gases (see Appendix A), under air flow, which most frequently yield a metal oxide (CuO for **1**) as final residue (R) (Table 4). TGA can provide information on melting, crystallization, sublimation, decomposition and solid state transitions, and also enable the observation and quantification of volatile compounds such as residual solvents and gaseous by-products. Some comments on TGA are as follows (Figure 4, left). It is widely documented that copper(II) compounds with uncoordinated water and aqua ligand(s) that satisfy distal position(s), with Cu-O > 3.32 Å, lose both in an overlapping fashion, in a global process, which starts below 100 °C but ends above this temperature. In contrast, the aqua ligands at proximal coordinating positions (Cu-O ~1.9–2.0 Å) are lost at significantly higher temperatures (usually >170 °C). In this context, compound **1** represents a fairly clear example. In the first stage, the loss of uncoordinated water (O3, see Figure 2) and the one that maintains a mere contact with copper(II) center (O2, also in Figure 2) are eliminated in an overall stage between 50–155 °C, with a difference of experimental and calculated weight loss of 0.813% (<1%). The second step involves the loss of the proximal aqua ligand (O1 in Figure 2 and Figure 3), but the loss of CO_2_ and the weight of the sample (6.387%) exceeds by 57.7% to the calculated value for the loss of the aqua ligand (4.049%). It seems obvious that in this step 2, a partial decarboxylation of tda occurs. In step 3, a reasonable agreement between the experimental and calculated weight loss is observed for the loss of tda. The evolved gases (in particular N_2_O in step 3 and SCNH in step 4) revealed that the burning of tda is also partially overlapped with that of 9heade. Additionally, the weight loss in stages 4 and 5 (30.390%) is lower than that estimated for the combustion of 9heade, leaving a CuO residue. It should also be noted that the final experimental residue (19.532%) is also somewhat higher than expected for the formation of only CuO (17.879%). The presence of S in tda and even more the abundance of N in 9heade allow us to speculate that the final residue (at 950 °C) contains some sulfate or most probably nitrate anions.

The differential scanning calorimetry of **1** (DSC, under N_2_ flow) shows two minima at 129 and 182 °C, respectively (Figure 4). These heat absorptions correspond to the losses of water weakly retained in the crystal (uncoordinated (O3) or giving a weak contact (O2) with the metal) and with the proximal aqua ligand (O1). The remaining recorded heat absorptions (up to 400 °C) cannot to be easily attributed to a melting with decomposition of the residue Cu(tda)(9heade), resulting in the absence of O_2_.

To confirm the bulk purity of the sample, the simulated and measured PXRDs were compared, and it was found that the two diffractograms are very similar, confirming that the crystalline bulk sample consists of a single phase (Appendix A). 

Likely, the initial sample used for the TGA experiment contained slightly less than two molecules of uncoordinated water, something assumable given the high environmental temperatures registered in our laboratories during the present work. In this connection, the room temperature ESR spectra, recorded on powdered samples, showed contributions from two magnetically independent Cu(II) species (Appendix A). The more intense one (~94%) is characterized by the following principal components of the g tensor: g_1_ = 2.304, g_2_ = 2.083 and g_3_ = 2.055 (<g> = 2.147; G = 4.4). These values imply a d_x_^2^ − _y_^2^ ground state and agree with the 4 + 1 coordination around copper(II), as reported by the crystallographic results. The secondary signal is very narrow and characteristic of an axially symmetric g tensor (g_//_ = 2.204; g_⊥_ = 2.140; <g> = 2.161; G = 1.5). Interestingly, the intensity of this line increases notably when the sample is ground, even if it is conducted very gently, suggesting that the new signal is originated by the loss of some of the water molecules of the compound. That is, the loss of these molecules modifies the magnetic interactions in the compound, as confirmed by the difference between the calculated G values for both signals [34]. Taking into account the two water-mediated H-bonding interactions, as referred above, such behavior can be rationalized assuming that a partial loss of any of the two uncoordinated water molecules can occur and provoke the formation of a secondary phase, according to our ESR results.

### 2.5. DFT Calculations

The DFT study is primarily focused on analyzing some assemblies observed in the solid state of compound **1**. Figure 5a shows a self-assembled dimer formed in the solid state of **1** via the Hoogsteen side of the adenine. This is the only possibility of self-assembly because the Watson–Crick side is blocked by the coordination of the adenine to Cu(II) through N1. The formation of the two symmetrically equivalent N6–H6⋯N7 H-bonds generates the *R*_2_^2^ (10) supramolecular ring. In addition, Figure 5b shows the interaction of one of the non-coordinated water molecules, which establishes a strong H-bond with the 2-hydroxyethyl group in such a way that one lone-pair (lp) points to the five membered ring of adenine, thus establishing a lp–π interaction (O⋯Cg distance: 3.38 Å). Both assemblies have been theoretically analyzed in this section.

The MEP surface of compound **1** was initially computed (see Figure 6) to investigate the most electron rich and electron poor regions. The MEP minimum is located at the non-coordinate O-atom of the carboxylate group (−63.5 kcal/mol). The MEP values are also negative at the N3 and N7 atoms of the adenine ring (−9 and −15 kcal/mol, respectively) and hydroxyl O-atom (−17 kcal/mol). The maximum MEP is located at H-atoms of the coordinated water molecule (+75 kcal/mol) as expected due to the strong effect of the coordination to Cu(II) on the acidity of the water protons. The MEP values are also large and positive at the H-atoms of the hydroxyl and amino groups, ranging from +53 to +57 kcal/mol. Moreover, the MEPs over the five- and six-membered rings of adenine are positive (29 and +28 kcal/mol, respectively), revealing that the π-surface is adequate for interacting with electron rich atoms. This MEP analysis also shows that the MEP is positive at the extension of the S-C bonds (+34 kcal/mol, σ-hole), evidencing that the coordinated S-atom of the tda ligand is able to act as Lewis acid and establish chalcogen bonds.

Figure 7a shows the combined QTAIM/NCIPlot analysis of the self-assembled H-bonded dimer commented above, evidencing that each H-bond is characterized by a bond critical point (CP, magenta sphere) and bond path (dashed bonds) interconnecting the H- and N-atoms. The N-H··N interactions are also characterized by blue (attractive) NCIplot isosurfaces coincident with the location of the bond CPs. The total dimerization energy is ΔE_1_ = −9.2 kcal/mol (−4.6 kcal/mol each H-bond), thus confirming their energetic relevance. Figure 7b shows the combined QTAIM/NCIPlot analysis of the water–adenine complex. The O-H⋯O interaction is characterized by a bond CP, bond path and dark blue reduced density gradient (RDG) isosurface interconnecting with the H- and O-atoms. Interestingly, the analysis also shows one bond CP and bond path connecting the O-atom of water to one C–atom of adenine, thus confirming the existence of the lp-π interaction. This contact is also disclosed by the NCIPlot analysis that shows a green and extended RDG isosurface located between the O-atom and the five-membered ring. The total interaction energy is moderately strong, ΔE_2_ = −5.9 kcal/mol. To estimate the contribution of the lp-π interaction, we have evaluated the strength of the O-H⋯O interaction by using the simple and reliable methodology recently proposed by Emaniam et al. [35]. In particular, the hydrogen bond strength (ΔE) is calculated using the electron density (ρ) at the bond CP and the equation ΔE = −233.1 × ρ + 0.7. The hydrogen bond strength is indicated in Figure 7b (in red close to the bond CP) revealing that the OH⋯O interaction is −4.7 kcal/mol, and consequently, the contribution of the lp-π is only −1.2 kcal/mol. A different methodology has also been used to estimate the lp-π interaction energy, consisting of rotating the C–C bond of the 2-hydroxyethyl group (maintaining the H-bonded water molecule) in such a way that the water molecule does not interact with the π-system. By comparing the energy of both rotamers, the lp–π interaction is estimated in 1.0 kcal/mol, which is similar to that deduced using the QTAIM method. Interestingly, the strength of the NH⋯N (Figure 7a) and OH⋯O interactions (Figure 7b) are similar in agreement with the color of the NCIplot RDG isosurfaces that characterize those H-bonds. In the lp–π complex (see Figure 5b and Figure 7b), the water is disposed parallel to the aromatic system. This type of arrangement has been described and studied in the literature by Zaric et al. [36]. In particular, by analyzing the CSD and using theoretical calculations, the authors revealed the existence of conformations where the water molecule is parallel to the aromatic ring plane at distances typical for stacking interactions. The attractive interaction energy reported by Zaric et al. [36] for benzene and water was −1.60 kcal/mol, which is consistent with the value reported herein.

In the solid state, compound **1** also forms infinite 1D assemblies where the electron rich O-atoms of the thiodiacetate coligand of one monomer are located approximately opposite to the C–S bonds of the adjacent monomer, thus propagating the supramolecular polymer (Figure 8). In addition, an OH⋯O H-bond between the coordinated water molecule and the carboxylate group is formed that further supports the propagation of the supramolecular polymer. It can be observed that one of both S⋯O contacts presents an intermolecular distance that is close (0.09 Å longer) to the sum of Van der Waals radii of O and S, ΣR_vdW_(O + S) = 3.32 Å. The distance of the other one is 0.3 Å longer than ΣR_vdW_(O + S); thus, the existence of such contact should be supported by theoretical calculations (vide infra).

Figure 9 shows the combined QTAIM/NCIPlot analysis of a dimer extracted from the 1D infinite chain. This analysis confirms the existence and attractive nature of both ChBs, each one characterized by a bond CP, bond path and green RDG isosurface connecting the S to the O-atom. The OH⋯O interaction is characterized by a bond CP, bond path and blue RDG isosurface coincident with the location of the bond CP. This reveals that the H-bond is significantly stronger than the ChBs. The QTAIM analysis also discloses the presence of ancillary interactions interconnecting both dimers, concretely two CH⋯O H-bonds and other contacts involving the π-systems (π⋯π, N⋯π and CH⋯π). Such an intricate combination of interactions justifies the large stabilization energy obtained for this assembly (ΔE_3_ = −29.5 kcal/mol).

At this point, it should be mentioned that the influence of water molecule’s coordination to a metal ion on the strength of hydrogen bonds has been previously studied both computationally and by analyzing the CSD, including the formation of OH⋯O interactions like those described herein [37].

## 3. Concluding Remarks

The molecular and crystal structures of the title compound revealed that in aqueous solution, the Cu(tda) chelate and the synthetic nucleoside 9heade recognize each other, forming a ternary assembly, which crystallizes as [Cu(tda)(9heda)(H_2_O)]·2H_2_O. The molecular recognition between the metal chelate and the adenine synthetic nucleoside represents a novel metal binding pattern (MBP) for 9heade, essentially bonded to Cu(II) by N1, its most basic donor atom available for coordination. Additionally, two water-mediated H-bonding interactions intra-stabilize the novel compound: 9heade-N6-H⋯O2(water) with O2⋯Cu weak contact and 9heade-O(ol)-H⋯O3(water) with O3-H36⋯π(centroid of the six-membered ring of adenine moiety). The easy partial loss of these water molecules makes it difficult to interpret the thermal stability and, above all, the spectral properties of the studied compound. However, both H-bonded and lp–π assemblies were analyzed energetically using density functional theory (DFT) calculations, reduced density gradient isosurfaces and the topological analysis of bond critical points, which was also used to estimate the contribution of the H-bonds. Additionally, structurally relevant S⋯O interactions between the Cu-coordinated S-atom and the carboxylate O-atoms of the tda ligand are rationalized by means of MEP and QTAIM calculations. These interactions are energetically relevant and dictate the X-ray packing of compound **1**.

## 4. Materials and Methods

### 4.1. Reagents and Synthesis of Compound ***1***

This product has been obtained by reaction between stoichiometric amounts of Cu_2_CO_3_(OH)_2_ (green malachite, Aldrich, Darmstadt, Germany), H_4_EDTA (Aldrich), thiodiacetic acid (TCI, Deutschland GmbH, Eschborn, Germany) and 9heade (TCI) in water. Various experiments were carried out. In a typical experiment, greenish malachite Cu_2_CO_3_(OH)_2_ (1 mmol, 0.22 g) and H_2_tda (2 mmol, 0.130 g) were reacted in 200 mL of distilled water, inside a covered Kitasato flask of 500 mL. Its open side outlet allows removing CO_2_ (the only by-product) and prevents possible splashes. The reaction mixture is continuously heated (45–50 °C) and stirred for an hour until unreacted malachite is not observed. The light green aqueous solution of Cu(tda) chelate was cooled to r.t. and slowly filtered by 250 mL additional glass Büchner funnel with sintered borosilicate plate (No. 3, porosity 16–40 µm) over an Erlenmeyer flask (caution: insufficient reaction time should be avoided since it leaves unreacted green malachite on the Büchner plate). Then, 9heade (0.36 g, 2 mmol) is added to the chelate solution, and the mixture stirred until their solution has been completed. The mother liquor of the ternary system was again filtered, over a proper Büchner and a 250 mL crystallization flask, being covered with a perforated plastic film to control the evaporation of the solvent. Working in this way, crystals of the desired product **1** appear in approximately three weeks. Handily picked single crystals were used for crystallographic purposes. Time-by-time samples were checked using FT-IR spectroscopy that revealed that only one product seems to be obtained. The product was dried at r.t. Overall yields are ca. 60–75%. Long standing of samples over 30 °C loses variable amounts of uncoordinated water, causing the problems referred in detail in the discussion of their physical properties. This also affects variable results of the CNH elemental analysis. Best agreement is as follow: for elemental analysis (%), calc. for C_11_H_19_CuN_5_O_8_S, C 29.70, H 4.30, N 15.74 and found, C 29.83, H, 4.18, N, 15.80.

### 4.2. Physical Measurements

The elemental analysis was performed with a Thermo Scientific Flash 2000 (Thermo Fisher Scientific Inc., Waltham, MA, USA). Infrared spectra (samples in KBr pellets) were recorded using a Jasco FT-IR 6300 spectrometer (Jasco Analítica, Madrid, Spain). Electronic (diffuse reflectance) spectra were obtained in a Varian Cary-5E spectrophotometer (Agilent Scientific Instruments, Santa Clara, CA, USA) from a ground crystalline sample. Thermo-gravimetric analyses (TGA) was carried out (10 °C/min) under air-dry flow (100 mL/min) with a thermobalance Mettler-Toledo TGA/DSC1 (Mettler-Toledo, Columbus, OH, USA), and a series of 50 time-spaced FT-IR spectra were recorded to identify evolved gases throughout the experiment, using a coupled FT-IR Nicolet 550 spectrometer (Thermo Fisher Scientific Inc., Waltham, MA, USA). Differential scanning calorimetry (DSC) measurement was recorded for a sample (6.234 mg) of compound **1**, on a DSC-SHIMADZU mod. DSC-50Q instrument (Shimadzu Europe, F.R. Germany GbmH) under an N_2_ atmosphere, at 30–400 °C (heating rate 10 °C/min). Powder X-ray diffraction (PXRD) data were collected using a BrukerD8 Advance Vario diffractometer (Bruker GmbH, Karlsruhe, Germany) with a Bragg Brentano parafocusing geometry and Cu Kα1 radiation (1.5406 Å). The tube voltage and amperage were set at 40 kV and 30 mA, respectively. The sample was scanned between 2 and 70° 2θ with a step size of 0.02°. The instrument was calibrated using a silicon standard prior to measurements. To preserve the sample, slurry was prepared, carefully crushing some crystals in their mother liquor, and measuring the slurry directly. Electron paramagnetic resonance (EPR) measurements were performed using a Bruker ELEXSYS E500 spectrometer operating at the X-band. The spectrometer was equipped with a super-high-Q resonator ER-4123-SHQ. The magnetic field was calibrated by a NMR probe and the frequency inside the cavity (~9.4 GHz) was determined with an integrated MW-frequency counter. Data was collected and processed using the Bruker Xepr suite.

### 4.3. Crystallography

A green block crystal of [Cu(tda)(9heade)(H_2_O)]·2H_2_O (**1**) was mounted on a glass fiber and used for data collection. Crystal data were collected at 298(2) K, using a Bruker D8 VENTURE diffractometer. Graphite monochromatic CuK(α) radiation (λ = 1.54184 Å) was used throughout. The data were processed with APEX3 [38] and corrected for absorption using SADABS (transmissions factors: 1.000–0.708) [39]. The structure was solved by direct methods using the program XT [40] and refined by full-matrix least-squares techniques against F^2^ using XL [40]. Positional and anisotropic atomic displacement parameters were refined for all non-hydrogen atoms. Hydrogen atoms were located in difference maps and included as fixed contributions riding on attached atoms with isotropic thermal parameters 1.2/1.5 times those of their carrier atoms. Criteria of a satisfactory complete analysis were the ratios of root mean squares shift in standard deviation of less than 0.001 and no significant features in final difference maps. Atomic scattering factors were obtained from the International Tables for Crystallography [41]. Molecular graphics were plotted with PLATON [42]. A summary of the crystal data, experimental details and refinement results are listed in Table 1.

### 4.4. Computational Details 

The calculations of the non-covalent interactions were carried out using Gaussian-16 [43] and the PBE0-D3/def2-TZVP level of theory [44,45]. To evaluate the interactions in the solid state, the crystallographic coordinates have been used. The interaction energies have been computed by calculating the difference between the energies of isolated monomers and their assembly. The interaction energies were calculated with correction for the basis set superposition error (BSSE) by using the Boys–Bernardi counterpoise technique [46]. The Bader’s “Atoms in molecules” theory (QTAIM) [47] has been used to study the interactions discussed herein by means of the AIMAll calculation package [48]. The molecular electrostatic potential surfaces (isosurface 0.001 a.u.) have been computed using the Gaussian-16 software [43]. The MEP surfaces were visualized using Gaussview 6.0 [49] program and the NCIPlot surfaces using the AIMAll program [48].

In order to assess the nature of interactions in terms of being attractive or repulsive and revealing them in real space, we have used NCIPLOT index, which is a method for plotting non-covalent interaction regions [50] based on the NCI (Non-Covalent Interactions) visualization index derived from the electronic density [50,51] The reduced density gradient (RDG), coming from the density and its first derivative, is plotted as a function of the density (mapped as isosurfaces) over the molecule of interest. The sign of the second Hessian eigenvalue times the electron density (i.e., sign(λ_2_)ρ in atomic units) enables the identification of attractive/stabilizing (blue–green colored isosurfaces) or repulsive (yellow–red colored isosurfaces) interactions using 3D-Plots. For the plots shown in Figure 7 and Figure 9, the NCIplot index parameters are RGD = 0.5, ρ cut off = 0.04 a.u., and color range: −0.035 a.u. ≤ sign(λ_2_)ρ ≤ 0.035 a.u.

## Data Availability

Not applicable.

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
