# Peer review of "The Copper(II)-Thiodiacetate (tda) Chelate as Efficient Receptor of N9-(2-Hydroxyethyl)Adenine (9heade): Synthesis, Molecular and Crystal Structures, Physical Properties and DFT Calculations of [Cu(tda)(9heade)(H2O)]·2H2O"

_molecules, 2023, doi:10.3390/molecules28155830_

Round 1

Reviewer 1 Report

This paper by Niclós-Gutiérrez and coworkers on the copper coordination compound containing synthetic nucleoside is a very interesting and detailed experimental and theoretical work. Indeed, one can wish that all inorganic synthetic chemists joined forces with computational chemists and performed this kind of work on their compounds in order to understand the bonding in coordination compounds (both covalent and noncovalent), which is an exciting area of research, with so many things yet to be discovered.

In spite of only dealing with one product compound, this paper deserves to be considered for publication in Molecules. However, before being accepted for publication, this paper has to address several questions.

- The CIF of the title compound is not provided with the manuscript, which is something I believe should be mandatory when submitting papers on new crystal structures. The CIF would be very helpful for the reviewers in order to avoid confusion in the structures with as many noncovalent interactions as in this one. For example, are π···π stacking interactions presented in Figure 9 significantly parallel-displaced, since they are combined with N···π and C-H···π interactions? I strongly encourage the authors to provide the CIFs in their future submissions, as well as the editors to make this a mandatory step in the process of manuscript submission.

- I suggest that the authors make a table in the introduction section with nine metal complexes having 9heade as ligand, instead of having the paragraph which is very difficult to read and even more difficult to understand in terms of their classification into several groups. Also, in this paragraph the full name of acac is misspelled and its charge is wrong (it is 1-, not 2-).

- It would be very helpful if the authors added a scheme of the synthesis of the title compound, including the structural formula. It can be added in the section 2.1., which has a very nice title “About the strategy of synthesis”.

- Atom labeling in Figure 3 caption does not correspond to atom labeling on the Figure 3 itself. I think the authors are trying to label noncovalent interactions in this figure according to the labeling of atoms in the Scheme 1, and this can be very confusing for the readers. The labeling of atoms should be the same on the figure and in the figure caption.

- The quality of Figure 4 is very poor and it should be improved.

- N-H···O interaction is mentioned on the page 10, line 337, and they are said to be presented on Figure 7-a. However, I do not see any N-H···O interactions in this figure. Did the authors actually mean N-H···N interactions?

- A question also arises regarding the true nature of interaction which authors denote as lp-π. From all the figures it looks like the entire water molecule is parallel with aromatic ring plane, which then corresponds to parallel water-aromatic interactions mentioned in Chem. Commun. 2008, 6546-6548. Even the calculated energy of this interaction (-1.2 kcal/mol) resembles the energies of parallel water-aromatic interactions mentioned in that paper. A more thorough discussion of this interaction would be appreciated.

- The authors show on Figure 9 that coordinated water forms C-H···O interaction with nucleic base, as well as O-H···O hydrogen bond. These interactions are known in the literature, their geometries and energies are described, and they should therefore be discussed and referenced in this paper.

- I find the last paragraph of DFT calculations section a bit confusing (lines 375-385). This paragraph should be rewritten.

- In Computational details section there is a mention of NCI plots in Figures 3 and 4. However, these figures do not contain any such plots. Please correct this.

- At several places there are wrong references. For example, on page 10, row 349, the methodology by Emaniam is mentioned, but it is referenced under the number 35, which is actually related to Bruker software. Also, I must question the validity of several references in the Computational details section, since reference 46 is not Gaussian-16, and I also do not think that reference 48 is related to NCI index, as mentioned in the text. Moreover, reference 47 is never mentioned in the text. The authors should carefully check all the references.

I find the quality of English language quite satisfactory. 

Reviewer 2 Report

In this manuscript, the authors present the synthesis and the experimental and theoretical characterization of copper-adenine-thidiacetate complexes. In particular, they analyze the non covalent interactions at play in the solid states using X-Ray structures and DFT computations.

The synthesis and experimental characterization are nicely conducted and presented. The theoretical analysis is correct but might benefit from some clarifications before publication.

I suggest that the authors address the following minor comments:

-   p10 line 348, the work of Emaniam et al. is presented in ref 48 and not 35.

-   p10 Figure 7: Another way to estimate the lp-π interaction would be do “suppress” it by turning the OH---OH2 fragment by 180° (on the arm linking it to the adenine moiety). This would provide a structure in which there is no more this lp-π interaction, and thus give another estimation of its energy.

-     -     p11 Figure 8 or 9: the authors should indicate the distance between the aromatic moieties, and may be the angle between the planes. This whould help characterizing this π - π interaction.

-   p13 “Computational details”. The authors should check the references. Indeed Gaussian16 line 482 is not ref 46 but ref 40. NCIPlot is not ref 48 but reef 46+47…

-   last, which software has been used to visualize the MEP and NCI surfaces?

Reviewer 3 Report

1.  The author needs to highlight the novel of work in abstract

2. There should be a new paragraph in this review article based on compares the findings of the most recent study.

3. There are many things wrong with English writing. The author needs to revise the manuscript thoroughly

4. Some of the work on the DFT could be updated, such as Acta Phys. -Chim. Sin. 2018, 34 (3), 303–313; Monatsh. Chem, 2019, 150, 1355–1364; Monatsh Chem, 2017, 48,1259–1267; J. Phys. Chem. A, 2019, 123, 6751−6760 and Theor. Chem. Acc. 2022, 141, 68

5. The Fig. 4 is badly, pls provide a new one

6. Pls removed the Table 1 into ESI

7. Pls provide the PXRD.

8. As to the TGA, pls explain it in detail.

see the comment

Round 2

Reviewer 1 Report

The authors have significantly improved the manuscript. However, the Table 1 in the introduction did not improve this part of the manuscript as much as it could have. The paragraph including the lines 69-80 gives us all the MBPs in a textual form, and I also think this could be presented in a more clear way by putting in into the table. After that, it should be better connected with the following paragraph and Table 1.

In summary, the contents of the paragraphs that include lines 69-90 should be presented in table(s), and the table should be briefly discussed in the text. After this is done, I fully support the acceptance and publication of this manuscript. 

Reviewer 3 Report

the author still not reply the comments. Pls work it carefully.

work
